ecology, microbiology, evolution

*Plasmodium*, transmission, overdispersion, temporal heterogeneity, avian malaria

**Author for correspondence:**
R. Pigeault
e-mail: romain.pigeault@univ-poitiers.fr

†Present address: Laboratoire EBI, Equipe EES, UMR CNRS 7267, Poitiers, France.

# Last-come, best served? Mosquito biting order and *Plasmodium* transmission

J. Isaïa[1], A. Rivero[2,3], O. Glaizot[1,4], P. Christe[1] and R. Pigeault[1,†]

[1]Department of Ecology and Evolution, University of Lausanne, CH-1015 Lausanne, Switzerland
[2]MIVEGEC (UMR CNRS 5290), Montpellier, France
[3]CREES (Centre de Recherche en Ecologie et Evolution de la Santé), Montpellier, France
[4]Musée Cantonal de Zoologie, Lausanne, Switzerland

 JI, 0000-0001-7737-9133; AR, 0000-0002-7056-5846; OG, 0000-0001-9116-3355; PC, 0000-0002-8605-7002; RP, 0000-0002-8011-4600

A pervasive characteristic of parasite infections is their tendency to be over-dispersed. Understanding the mechanisms underlying this overdispersed distribution is of key importance as it may impact the transmission dynamics of the pathogen. Although multiple factors ranging from environmental stochasticity to inter-individual heterogeneity may explain parasite overdispersion, parasite infection is also overdispersed in an inbred host population maintained under laboratory conditions, suggesting that other mechanisms are at play. Here, we show that the aggregated distribution of malaria parasites within mosquito vectors is partially explained by a temporal heterogeneity in parasite infectivity triggered by the bites of mosquitoes. Parasite transmission tripled between the mosquito's first and last blood feed in a period of only 3 h. Surprisingly, the increase in transmission is not associated with an increase in parasite investment in production of the transmissible stage. Overall, we highlight that *Plasmodium* is capable of responding to the bites of mosquitoes to increase its own transmission at a much faster pace than initially thought and that this is partly responsible for overdispersed distribution of infection. We discuss the underlying mechanisms as well as the broader implications of this plastic response for the epidemiology of malaria.

## 1. Introduction

A ubiquitous feature of parasite infections is their tendency to be overdispersed [1–4]. In other words, in a natural population of hosts, the majority of individuals tend to harbour few or no parasites, while a few hosts harbour the vast majority of the parasite population. This pattern has been observed in a wide range of diseases, from viruses and fungal parasites of plants [5,6] to protozoan and metazoan parasites of humans [7,8].

Previous work has shown that the overdispersed pattern of parasites among hosts can have important consequences for disease dynamics [9,10]. Overdispersion reduces the deleterious effects of parasites on host populations and increases the intensity of density-dependent suppression of parasite population growth (e.g. mating probability, intra- and inter-specific competition [11,12]). Another property emerging from parasite overdispersion is the effect on parasite transmission. The small fraction of heavily infected individuals may act as super-spreaders, playing a large role in disease transmission [13–15]. In many host–parasite systems, 20% of hosts are responsible for 80% of new infections [16,17]. In vector-borne diseases, parasite overdispersion has been observed both in vertebrate hosts and in vector populations [18–22]. Despite this, studies have mainly focused on the epidemiological consequences of parasite over-dispersion for the host, rather than for the vectors [17,23,24]. Yet, for many of these diseases, key traits determining the transmission dynamics of the

pathogen (e.g. lifespan, the length of the parasite's extrinsic incubation period) may depend on the intensity of parasite infection in the vector [25–29].

Anderson & Gordon identified environmental stochasticity as the prime cause of overdispersion in parasite populations [30]. Environmental stochasticity refers to both the physical parameters of the environment and the differences in host susceptibility resulting from behavioural differences, genetic factors or past experiences of infection. The mechanisms underlying the aggregated distribution of parasites in vector populations remain rarely explored and little understood.

*Plasmodium* parasites are known for being the aetiological agents of malaria and for their devastating effects on human populations. These vector-borne parasites also infect many other terrestrial vertebrate species, including other mammals, reptiles and birds. The life cycle of this parasite is the same in hosts of all taxa. When the mosquito vectors take a blood meal on an infected host, they ingest the parasite's transmissible stages (female and male gametocytes). After sexual reproduction of the parasite, the motile zygotes penetrate the wall of the mosquito midgut and start developing into oocysts, which in turn produce the sporozoites. These sporozoites, once in the mosquito's salivary glands, can be transmitted to a new vertebrate host. There is abundant evidence that the distribution of oocysts, the most commonly quantified parasite stage in mosquitoes, is highly overdispersed [8,31–33]. The simplest explanation for this aggregated distribution of oocysts is that vectors vary in susceptibility to *Plasmodium* infection according to their genetic background or to their physiological status [8,34,35]. Polymorphism in mosquito immune genes is strongly associated with natural resistance to *Plasmodium* [34,36], and ageing tends to decrease the susceptibility of vectors to *Plasmodium* infection [35]. Puzzlingly, however, oocyst overdispersion is also common under controlled laboratory conditions in highly inbred, and therefore physiologically and genetically homogeneous, mosquito populations [8,31,33]. This suggests that factors other than the genetic or physiological variations between mosquitoes may contribute to the aggregated distribution of oocysts in vectors.

Spatial aggregation of gametocytes in vertebrate blood could be partially responsible for the aggregated distribution of oocysts in mosquitoes. Recent work has shown that gametocyte density can change by more than 0.4-fold between blood collected from different human body parts ([37], but see [38]). Although the direct connection between spatial heterogeneity in vertebrate blood and overdispersion in mosquitoes has never been made, it has been reported that *Plasmodium* gametocytes show an aggregated distribution within mosquitoes that recently fed on a human host [39].

The aggregated distribution of *Plasmodium* parasites within mosquitoes could also be due to within-host *temporal* variation in parasite density and/or infectivity. Under this scenario, mosquitoes feeding during the high parasite density/infectivity phase would become more heavily infected than those feeding during the low density/infectivity phase. *Plasmodium* parasite density and/or infectivity in the vertebrate host can indeed vary over relatively short time scales. A recent study found that rodent malaria *Plasmodium chabaudi* gametocytes are twice as infective at night despite being less numerous in the blood [40]. A periodic late afternoon increase in parasitaemia is also observed in the avian malaria system [41]. Such temporal variation may be a function of changes in the physiological, nutritional or immunological condition of the host [42–44].

It may, however, also be an adaptive parasite strategy aimed at maximizing its own transmission [41,45]. Recent work has shown that host parasitaemia increases a few days after a mosquito blood feeding bout, suggesting that *Plasmodium* may be capable of adjusting its transmission strategy by responding plastically to the temporal fluctuations in vector availability [41,45]. These results, however, are not able to explain the aggregated distribution of parasites among mosquitoes feeding within a short feeding bout typically lasting a few hours.

Here, we test whether *Plasmodium* is able to respond plastically to the bites of mosquitoes at a much more rapid pace than initially thought. More specifically, we test whether there is a pattern in the oocyst load of mosquitoes feeding within a short (3 h) time interval: do the bites of the first mosquitoes increase the infectivity of the parasite such that mosquitoes biting later end up infected with more oocysts? To test this hypothesis, we use the avian malaria system, the only currently available animal experimental system that allows working with a parasite recently isolated from the wild (*Plasmodium relictum*), with its natural mosquito vector (*Culex pipiens* [46,47]). Specifically, we carry out a series of experiments designed to answer two main questions: (i) Is oocystaemia correlated with mosquito biting order? In other words, do mosquitoes biting first have a lower intensity of infection than those biting later on? and (ii) Is this due to a temporal increase in the parasitaemia/gametocytaemia in the birds as a result of mosquito bites?

## 2. Material and methods

### (a) Parasite and mosquito

Three experiments were carried out using three different isolates of *P. relictum* (lineage SGS1). The parasite strain used in the first block of the first experiment was isolated from an infected great tit (*Parus major*) in 2015. The strain used in the second experiment was isolated from an infected great tit in 2018. The strain used in the second block of the first experiment and in the third experiment was isolated from an infected house sparrow (*Passer domesticus*) in 2019. All strains were maintained through regular passages across our stock canaries (*Serinus canaria*) using intraperitoneal (i.p.) injections until the beginning of the experiment.

All experiments were conducted with *C. pipiens* mosquitoes collected in Lausanne (Switzerland) in August 2017, and maintained in the insectary since. Mosquitoes were reared using standard protocols [48]. We used females 7–13 days after emergence, which had not had prior access to blood. Mosquitoes were maintained on glucose solution (10%) since their emergence and were starved (but provided with water to prevent dehydration) for 24 h before the experiment.

### (b) Experimental design

To investigate the impact of mosquito bite-driven plasticity on *Plasmodium* transmission, three experiments were carried out in which infected birds (which had no previous exposure to haematophagous invertebrates) were exposed to mosquitoes for 3 h (18.00–21.00) and mosquitoes were sampled at regular intervals thereafter (different protocols for the three experiments, see below). To investigate the impact of vector bites on parasite population growth, the parasitaemia (number of parasites in the blood) and gametocytaemia (number of mature gametocytes in the blood) of vertebrate hosts exposed or not (control) to mosquitoes were measured just before and just after the

mosquito exposure period using blood smears [49] (electronic supplementary material, table S1). Although parasitaemia and gametocytaemia are highly correlated in this system (see fig. 2 in [46], and electronic supplementary material, table S1), we measured both variables in order to control for potential changes in conversion rates (density of gametocytes relative to the total number of parasites).

All experiments were carried out using domestic canaries (*S. canaria*). Birds were inoculated by intraperitoneal injection of 100 µl of an infected blood pool (day 0). The blood pool was made with a 1 : 1 mixture of phospate-buffered saline (PBS) and blood sampled from 2–4 canaries infected with the parasite three weeks before the experiment.

### (i) Experiment 1: oocyst burden and mosquito biting order: batch experiment

Two experimental blocks were carried out with two different isolates of *P. relictum*, using 14 and 5 infected birds, respectively. On days 11–13 post-infection, corresponding to the acute phase of infection, blood was sampled from each bird at 17.45. Straight after blood sampling, birds were individually placed in experimental cages (L 40 × W 40 × H 40 cm).

At 18.00, eight and three haphazardly chosen birds, from blocks 1 and 2, respectively, were exposed to four successive batches of 25 ± 3 uninfected female mosquitoes. Each mosquito batch was left in the cage for 45 min before being taken out and replaced by a new batch (i.e. batch 1 ($T_{0min}$), batch 2 ($T_{45min}$), batch 3 ($T_{90min}$) and batch 4 ($T_{135min}$)). Blood-fed mosquitoes from both batches were counted and individually placed in numbered plastic tubes (30 ml) covered with mesh and containing a cotton pad soaked in 10% glucose solution. At the end of the last mosquito exposure session (21.00), a second blood sample was taken from each bird. A red lamp was used to capture blood-fed mosquitoes without disturbing the birds and the mosquitoes. Control (unexposed) birds were placed in the same experimental conditions but without mosquitoes.

Tubes containing the blood-fed mosquitoes were kept in standard insectary conditions to obtain an estimate of the blood meal size and the level of infection (infection prevalence and oocyst burden). For this purpose, 7–8 days post blood meal, the females were taken out of the tubes and the amount of haematin excreted at the bottom of each tube was quantified as an estimate of the blood meal size [48]. Females were then dissected, and the number of *Plasmodium* oocysts in their midgut counted with the aid of a binocular microscope [48].

### (ii) Experiment 2: oocyst burden and mosquito biting order: individual monitoring

To obtain a finer measurement of the impact of mosquito biting order on oocyst burden, a second experiment was carried out with the same protocol as described above, except that the birds (four of the eight infected birds) were exposed to a single batch of 100 uninfected mosquitoes for 3 h (18.00–21.00). Female mosquitoes were continuously observed and individually removed from the cages immediately after blood feeding in order to record the order of biting of each female.

### (iii) Experiment 3: mosquito biting order and density of parasites ingested

A third experiment was carried out to investigate whether the total number of parasites in the blood meal, immediately after the blood feeding, fluctuated during the feeding bout. Two infected birds were individually placed in experimental cages and exposed to a single batch of 100 mosquitoes for 3 h (18.00–21.00). Mosquitoes were removed from the cages immediately after blood feeding. The order of biting of each female was recorded and every

second mosquito collected was either immersed immediately in liquid nitrogen to quantify the number of parasites ingested by quantitative polymerase chain reaction (qPCR) or stored in an individual plastic tube and dissected one week later to count the number of oocysts in the midgut.

### (c) Molecular analyses

The quantification of parasites contained within the blood meal was carried out using qPCR with a protocol adapted from [50]. Briefly, DNA was extracted from blood-fed females using standard protocols (Qiagen DNeasy 96 blood and tissue kit). For each individual, we conducted two qPCRs: one targeting the nuclear 18S ribosomal DNA (rDNA) gene of *Plasmodium* (primers 18sPlasm7 5'-AGCCTGAGAAATAGCTACCACATCTA-3', 18sPlasm8 5'-TG TTATTTCTTGTCACTACCTCTCTTCTTT-3') and the other targeting the 18S rDNA gene of the bird (primers 18sAv7 5'-GAA ACTCGCAATGGCTCATTAAATC-3', 18sAv8 5'-TATTAGCTCTA GAATTACCACAGTTATCCA-3'). All samples were run in triplicate (Bio-Rad CFX96™ Real-Time System). Samples with a threshold $C_t$ value higher than 35 were considered uninfected. Parasite number was calculated with relative quantification values (RQ). RQ can be interpreted as the fold-amount of the target gene (*Plasmodium* 18S rDNA) with respect to the amount of the reference gene (bird 18S rDNA) and are calculated as $2^{-(C_t 18S\ Plasmodium\ -\ C_t 18s\ Bird)}$. For convenience, RQ values were standardized by $\times 10^4$ factor and log-transformed.

### (d) Statistical analyses

Analyses were carried out using the R statistical package (v. 3.4.1). Data were analysed separately for each experiment and each experimental block. Blood meal rate, blood meal size, infection prevalence, oocyst burden (where only individuals that developed ≥ 1 oocyst were included) and quantity of parasites contained within the blood meal (which may depend on which bird mosquitoes fed on) were analysed, fitting bird as a random factor into the models using *lmer*, *glmer* or *glmer.nb* (package: lme4 [51]) according to whether the errors were normally (haematin quantity, and quantity of parasites contained within the blood meal), binomially (blood meal rate and infection prevalence) or negative binomially distributed (oocyst burden). Blood meal size (when it was not a response variable) and mosquito batches (experiment 1) or mosquito biting order (experiments 2 and 3) were used as fixed factors. Parasitaemia and gametocytaemia of birds were analysed using *lmer* with bird fitted as a random factor into the models to account for temporal pseudo-replication. Times of day (17.45 and 21.00) and bird group (exposed to mosquito bites or control) were used as fixed factors.

The different statistical models built to analyse the data are described in electronic supplementary material, table S2. Maximal models, including all higher-order interactions, were simplified by sequentially eliminating non-significant terms and interactions to establish a minimal model [52]. The significance of the explanatory variables was established using a likelihood ratio test [53]. The significant chi-squared values given in the text are for the minimal model, whereas non-significant values correspond to those obtained before the deletion of the variable from the model. *A posteriori* contrasts were carried out by aggregating factor levels and by testing the fit of the simplified model using a likelihood ratio test [52].

## 3. Results

### (a) Experiment 1: oocyst burden and mosquito biting order: batch experiment

In this experiment, birds were exposed to four successive batches of 25 ± 3 uninfected mosquitoes. Each mosquito

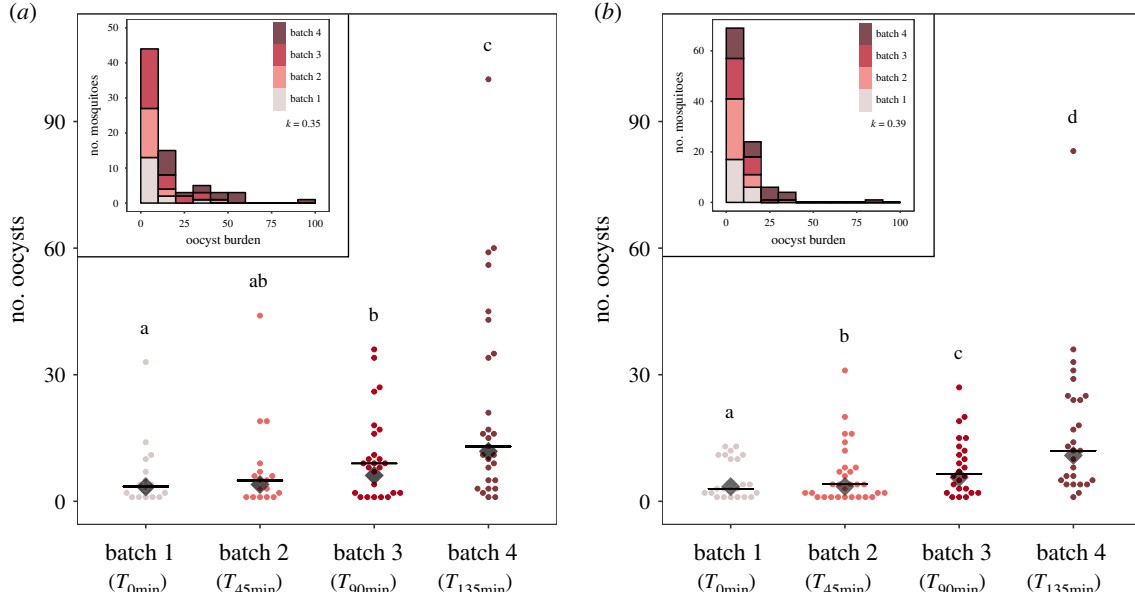

**Figure 1.** Experiment 1: impact of mosquito batch order on *Plasmodium* transmission. Number of oocysts in the midgut of *Plasmodium*-infected mosquitoes according to mosquito batch. Each mosquito batch was left in contact with birds for 45 min (batch 1 ($T_{0min}$), batch 2 ($T_{45min}$), batch 3 ($T_{90min}$) and batch 4 ($T_{135min}$)). Birds were infected either by a *P. relictum* lab strain (experimental block 1, *a*) or by a *P. relictum* strain freshly collected in the field (experimental block 2, *b*). Black horizontal lines represent medians and black diamonds represent geometric means. Levels not connected by the same letter are significantly different. Histograms in each panel show the distribution of oocyst burden in mosquitoes in the experimental blocks 1 (*a*) and 2 (*b*); the colours represent the mosquito batches (from 1 to 4). *k*, clumping parameter. (Online version in colour.)

batch was kept in the cage for 45 min before being replaced with a new batch (batch 1 ($T_{0min}$), batch 2 ($T_{45min}$), batch 3 ($T_{90min}$) and batch 4 ($T_{135min}$)). The blood meal rate (i.e. proportion of blood-fed mosquitoes) and the haematin quantity, a proxy for blood meal size, were similar for all batches (mean ± s.e., blood meal rate: batch 1: 19% ± 6, batch 2: 23% ± 8, batch 3: 29% ± 4 and batch 4: 31% ± 3, model 1: $\chi^2 = 5.90$, $p = 0.116$; haematin excreted (ng): batch 1: 17.58 ± 1.60, batch 2: 18 ± 1.76, batch 3: 17.34 ± 1.54 and batch 4: 18.29 ± 1.8, model 2: $\chi^2 = 3.55$, $p = 0.314$). Although mosquitoes from batches 3 and 4 tended to have a higher infection prevalence (proportion of mosquitoes containing at least one oocyst in the midgut; batch 3: 64.4% ± 11.9 and batch 4: 78.2% ± 8.6) than those from batches 1 and 2 (batch 1: 56.7% ± 15 and batch 2: 56.7% ± 19.4), the difference in prevalence between the different batches was not statistically significant (model 3: $\chi^2 = 2.74$, $p = 0.433$). The overall distribution of oocyst burden across batches was highly overdispersed (figure 1*a*; mean ± s.e., variance-to-mean ratio (VMR) = 11.48 ± 3.37, clumping parameter $k = 0.35$ [52]). Oocyst burden increased with mosquito batch (geometric mean: batch 1: 3.41 ± 3.04, batch 2: 3.99 ± 3.25, batch 3: 6.13 ± 3.36 and batch 4: 11.84 ± 3.53, model 4: $\chi^2 = 35.283$, $p < 0.0001$; figure 1*a*). Females from batch 4 had almost twice as many oocysts as those from batch 3 (contrast analyses: batch 4/batch 3: $\chi^2 = 11.02$, $p < 0.001$) and three times more than females from batches 1 and 2 (batch 4/batch 2: $\chi^2 = 17.95$, $p < 0.001$, batch 4/batch 1: $\chi^2 = 19.31$, $p < 0.0001$; figure 1*a*). No significant difference was, however, observed between mosquitoes from batches 1 and 2 (contrast analyses: batch 1/batch 2: $\chi^2 = 0.15$, $p = 0.697$) or between mosquitoes from batches 2 and 3 (batch 2/batch 3: $\chi^2 = 2.29$, $p = 0.129$; figure 1*a*). When the analysis was re-run removing outliers (threshold: third quartile + 1.5 × interquartile range), the mosquito biting order still had a significant effect on the oocyst burden (model 5: $\chi^2 = 12.43$, $p = 0.006$, geometric mean without outliers: batch 1: 3.41 ± 3.04, batch 2:

3.43 ± 2.82, batch 3: 6.13 ± 3.36, batch 4: 7.52 ± 2.82). Haematin quantity exhibited no significant association with the oocyst burden (model 4: $\chi^2 = 0.02$, $p = 0.875$).

The increase in *Plasmodium* oocyst burden with mosquito batch was not explained by an increase in total parasite or gametocyte burden in the birds' peripheral blood. The parasitaemia and gametocytaemia of exposed birds remained roughly constant throughout the experiment (parasitaemia: model 6: $\chi^2 = 0.39$, $p = 0.529$; gametocytaemia: model 7: $\chi^2 = 0.02$, $p = 0.877$; electronic supplementary material, table S1) and were similar between exposed and unexposed (control) birds (parasitaemia: model 6: $\chi^2 = 0.29$, $p = 0.5907$; gametocytaemia: model 7: $\chi^2 = 0.60$, $p = 0.4364$; electronic supplementary material, table S1).

To test the repeatability of our results, a second experimental block, with a new *P. relictum* strain freshly collected in the field, was performed. The results of block 2 fully confirmed those of the first block. The blood meal rate and the quantity of haematin excreted by mosquitoes were similar for all batches (blood meal rate: batch 1: 50% ± 11, batch 2: 45% ± 17, batch 3: 38% ± 3 and batch 4: 43% ± 3, model 8: $\chi^2 = 1.77$, $p = 0.621$; haematin excreted (ng): batch 1: 24.78 ± 1.46, batch 2: 26.43 ± 1.66, batch 3: 24.88 ± 1.96 and batch 4: 26.65 ± 1.42, model 9: $\chi^2 = 1.14$, $p = 0.766$). The difference in infection prevalence between the different batches was not statistically significant (model 10: $\chi^2 = 5.64$, $p = 0.130$) although mosquitoes from batches 2, 3 and 4 tended to have a higher prevalence (mean ± s.e., batch 2: 73.1% ± 7.0, batch 3: 68.6% ± 7.9 9 and batch 4: 71.1% ± 7.5) than those from batch 1 (mean ± s.e., batch 1: 51.1% ± 7.5). The distribution of oocyst burden in mosquitoes was overdispersed (figure 1*b*; mean ± s.e., VMR = 11.40 ± 5.66, $k = 0.39$) and we observed a significant increase in oocyst burden with mosquito batch order (model 11: $\chi^2 = 34.34$, $p < 0.0001$; geometric mean: batch 1: 3.48 ± 2.69, batch 2: 3.52 ± 2.95, batch 3: 5.63 ± 2.72 and batch 4: 10.87 ± 2.75, all contrast analyses were significant; figure 1*b*). When the analysis was re-run by

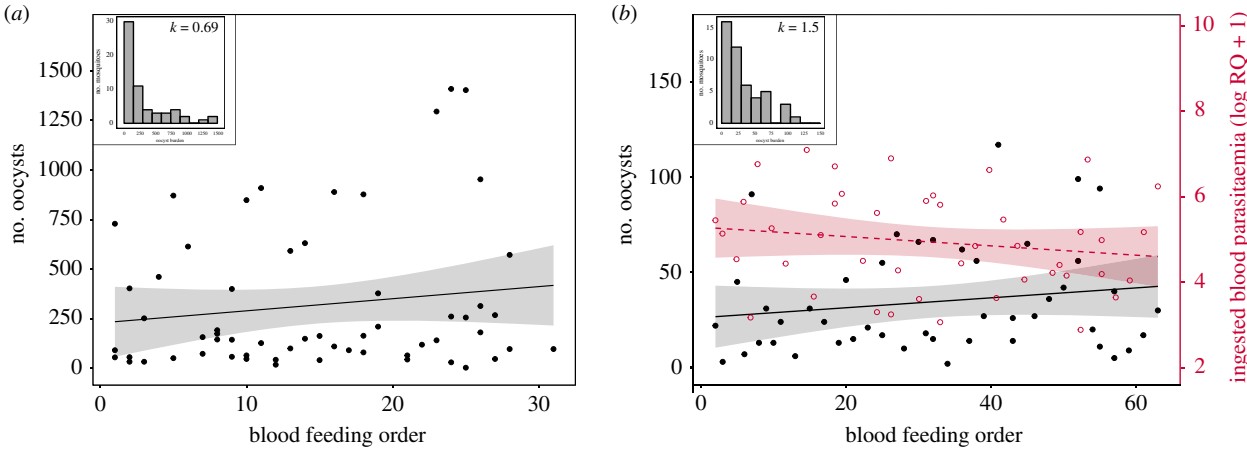

**Figure 2.** Effect of individual mosquito blood feeding order on the number of parasites ingested and on the intensity of infection. (*a*) Relationship between oocyst burden and mosquito biting order (experiment 2). (*b*) Relationship between the number of parasites ingested (red open dots and red dashed line), or the oocyst burden (black dots and black solid line), and the mosquito biting order (experiment 3). Each point represents one blood-fed mosquito. Shaded areas on either side of the regression line represent 95% confidence intervals. Histograms show the distribution of oocyst burden in mosquitoes in experiment 2 (*a*) and 3 (*b*). k, clumping parameter. (Online version in colour.)

removing outliers, the mosquito biting order still had a significant effect on the oocyst burden (model 12: $\chi^2 = 19.307$, $p = 0.0002$; geometric mean without outliers: batch 1: $3.48 \pm 2.69$, batch 2: $3.26 \pm 2.77$, batch 3: $5.63 \pm 2.72$ and batch 4: $8.13 \pm 2.39$). A significant positive correlation between haematin and oocyst burden was found (model 11: $\chi^2 = 4.46$, $p = 0.03$). As in the previous experimental block, the vertebrate host parasitaemia and gametocytaemia remained constant throughout the experiment (parasitaemia: model 13: $\chi^2 = 1.29$, $p = 0.256$; gametocytaemia: model 14: $\chi^2 = 0.88$, $p = 0.349$, respectively) and were similar between exposed and unexposed (control) birds (parasitaemia: model 13: $\chi^2 = 2.44$, $p = 0.118$; gametocytaemia: model 14: $\chi^2 = 2.45$, $p = 0.117$, respectively; electronic supplementary material, table S1).

## (b) Experiment 2: oocyst burden and mosquito biting order: individual monitoring

To obtain a finer measurement of the impact of mosquito biting order on parasite transmission, infected birds were exposed to 100 mosquitoes for 3 h (18.00–21.00) and mosquitoes were individually removed from the cages immediately after blood feeding. Haematin quantity and infection prevalence were independent of the mosquito biting order (haematin excreted (ng) = $24.91 \pm 1.26$, model 13: $\chi^2 = 2.44$, $p = 0.118$; infection prevalence = 98.6%, model 16: $\chi^2 = 0.83$, $p = 0.363$). The distribution of oocyst burdens across all mosquitoes was highly overdispersed (mean ± s.e., VMR = $90.26 \pm 41.53$, $k = 0.69$; figure 2*a*). Biting order was a significant explanatory factor of oocyst burden: mosquitoes that bit later showed higher oocyst burden than mosquitoes that bit earlier (model 17: $\chi^2 = 8.28$, $p = 0.004$; figure 2*a*). Haematin quantity and oocyst burden were significantly positively correlated (model 17: $\chi^2 = 19.151$, $p < 0.001$). As in the first experiment, vertebrate host parasitaemia and gametocytaemia remained constant throughout the experiment (parasitaemia: model 18: $\chi^2 = 2.03$, $p = 0.154$; gametocytaemia: model 19: $\chi^2 = 0.13$, $p = 0.718$, respectively) and were similar between exposed and unexposed (control) birds (parasitaemia: model 18: $\chi^2 = 0.98$, $p = 0.321$; gametocytaemia: model 19: $\chi^2 = 0.12$, $p = 0.731$; electronic supplementary material, table S1).

## (c) Experiment 3: number of parasites ingested and mosquito biting order

The first two experiments showed an increase in the oocyst burden with the order of mosquito bites but did not show a corresponding increase in parasite density in the peripheral blood of the vertebrate hosts (measured from blood samples). We carried out a third experiment to determine whether the total number of parasites in the blood meal fluctuated during the feeding bout. The amount of parasite ingested by the mosquitoes remained roughly constant throughout the exposure period (model 20: $\chi^2 = 1.54$, $p = 0.215$; figure 2*b*). The haematin quantity and the infection prevalence (oocyst stage) were also independent of the mosquito biting order (haematin excreted (ng) = $10.38 \pm 0.76$, model 21: $\chi^2 = 1.89$, $p = 0.169$; infection prevalence: 98%, model 22: $\chi^2 = 0.37$, $p = 0.545$). In contrast, the distribution of oocyst burden across all mosquitoes was still overdispersed (mean ± s.e., VMR = $15.03 \pm 1.86$, $k = 1.5$; figure 2*b*) and was significantly associated with mosquito biting order (model 23: $\chi^2 = 6.45$, $p = 0.011$; figure 2*b*). As in experiments 1 and 2, mosquitoes that bit later showed higher oocyst burden than mosquitoes that bit earlier (figure 2*b*). Haematin quantity had no effect on the oocyst burden (model 23: $\chi^2 = 3.77$, $p = 0.052$).

## 4. Discussion

Overdispersed distribution of vector-borne parasites in vertebrate and invertebrate host populations has important consequences for parasite transmission and disease control strategies [16,28,54]. Parasite overdispersion is driven by multiple factors ranging from population processes to inter-individual heterogeneity in susceptibility and parasite exposure [55–57]. Here, using three different isolates of *P. relictum*, we provide evidence that the aggregated distribution of malaria parasites within mosquito vectors may also be explained by mosquito biting order. On average, $10 \pm 3\%$ of the variation in oocyst burden was explained by biting order: mosquitoes that bite first end up with a lower intensity of infection than those that bite later on. This fluctuation in *Plasmodium* infectivity may reflect an adaptive strategy of parasites to optimize transmission.

The abundance of invertebrate vectors fluctuates at time scales ranging from days to years [40,58,59]. Previous studies have shown that malaria parasites have evolved two different and complementary transmission strategies to cope with both short- (circadian) and long-(seasonal) term fluctuations in mosquito activity. *Plasmodium* adopts an unconditional strategy whereby within-host parasitaemia and/or gametocyte infectivity varies daily, coinciding with the activity levels of its vector [41,42], but also a plastic strategy, allowing parasite growth to increase after exposure to mosquito bites [41,45,60]. This plastic strategy allows the parasite to react to daily and seasonal fluctuations in mosquito abundance by increasing its overall parasitaemia in the vertebrate blood [41,45].

In this study, we demonstrate that the plastic response of *Plasmodium* is much faster than initially thought [41,45]. When vertebrate hosts were exposed to mosquito bites during a short period of time (3 h), the parasite burden in mosquitoes increased gradually with their biting order. The density of parasites within the mosquito midgut tripled between the first and the last blood-fed mosquito. Although the biting order of the mosquitoes cannot be decoupled from the biting time (these two parameters are obviously highly correlated), the increase in the intensity of infection in such a short period of time suggests that the effect observed here cannot be explained solely by circadian fluctuation in parasite density in vertebrate blood. Many mosquito species exhibit a circadian rhythm in the host-biting activity but stochastic environmental factors such as variations in temperature, wind or humidity drastically affect the abundance of mosquitoes from one day to another [61–63]. Therefore, the association between an unconditional strategy (circadian fluctuation) and a quick plastic response to mosquito bites may allow malaria parasites to fine-tune investment in transmission according to the presence of mosquitoes.

Interestingly, this adaptive hypothesis involving an active parasite response to mosquito bites is not mediated by an increase in either parasite replication rate or gametocyte production: parasitaemia and gametocytaemia of birds exposed to mosquitoes were not different before and after mosquito probing. This result was confirmed by monitoring the number of parasites ingested by the mosquitoes immediately after the blood meal, throughout the exposure period. These results contrast with those obtained in recent studies [41,45], where the increase in oocyst burden observed in mosquitoes fed on a host a few days after the host was exposed to vector bites was correlated with an increase in parasitaemia and gametocytaemia. Our study suggests that malaria parasites may be employing an alternative strategy that allows them to react rapidly to the bites. One possibility is that *Plasmodium* may be reacting to mosquito bites by altering the physiological state of the gametocytes to render them more infectious. It was suggested as early as 1966 [64] that malaria parasite infectivity is not only due to the number of gametocytes in the blood but also to their physiological state. This prediction was recently experimentally confirmed by a study carried out with a rodent malaria parasite: *P. chabaudi* gametocytes were twice as infective at night despite being less numerous in the blood [40]. The mechanisms underlying gametocyte infectivity remain poorly understood. Although we know that gametocytes go through several stages (from one to eight depending on the species of *Plasmodium* [65]) of development before reaching the 'mature' stage, we do not know whether the mature

stage is systematically infectious. The mechanism by which malaria parasites accelerate the rate of maturation and/or infectivity of gametocytes in response to mosquito bites should be explored.

The response of the vertebrate host to mosquito bites could also enhance parasite transmission from the vertebrate host to the invertebrate vector by two non-exclusive mechanisms: (i) increased infectivity and/or survival of parasites in vector midgut and (ii) modified susceptibility of mosquitoes to infection. *Plasmodium* abundance varies drastically during its journey within the mosquito, which is partly intertwined with the kinetics of blood digestion [32]. Within seconds of ingestion by the mosquito, the drop in temperature and the rise in pH, associated with the presence of xanthurenic acid, trigger gametocyte activation and differentiation into gametes [64–66]. Studies on ookinete production have revealed that in addition to mosquito-derived xanthurenic acid, there are a series of undefined blood-derived factors ingested by mosquitoes that act as significant sources of gametocyte activation [67,68]. Indeed, numerous host blood-derived compounds remain or become active during mosquito blood digestion. Complement components, vertebrate antibodies and regulator factor H may impact gametocytes-to-zygote and zygote-to-ookinetes stages transition and survival [69–71]. Several studies also showed that ingested vertebrate-derived factors negatively impact mosquito microbiota (e.g. complement cascade [72]) and their peritrophic matrix (e.g. chitinase [71,73]), both of which are known to play a key role in the mosquito refractoriness to *Plasmodium* infection [74]. The concentration of these vertebrate-derived compounds in the ingested blood and, ultimately, their impact on parasite infectivity and/or vector susceptibility might progressively increase with the number of bites and thus explain the increase in oocyst density with mosquito biting order.

Our study did not determine whether the increase in oocyst burden with mosquito biting order was mediated by a plastic response of the parasite or by a response of the vertebrate host to mosquito exposure. Further work comparing the transcriptome of vertebrate hosts and parasites before and after mosquito exposure would be a useful first step. It would be also relevant to determine how biting order affects infection intensity with a membrane-feeding assay. This method would allow manipulative experiments with inactivation of the serum and cross-manipulation of infected and uninfected blood samples. Another important point that remains unknown is whether the progressive increase in parasite burden observed in mosquitoes was the result of a localized or a systemic phenomenon within the bird. If the gradual increase in parasite density was mediated by a local response of the parasite or vertebrate host, one would expect to observe this effect only among mosquitoes biting the same area of the body. Conversely, if the effect was due to a more general host response (e.g. increased blood levels of stress hormones [75]), then the relationship between mosquito biting order and parasite burden would be observed regardless of the body area bitten.

Here, we report a higher oocyst burden than is found in the field [76,77]. All experimental infections carried out in the laboratory (e.g. human and avian) produce oocyst burdens that are substantially larger than those found in the field because infections are carried out under optimal conditions (e.g. no host defensive behaviour, optimal temperature and humidity). We also report strong variation in oocyst burdens

between the different experiments. In experiment 2, the average parasite density in mosquitoes was more than 10 times higher than those observed in the other experiments. It was unlikely that these differences were due to variations in the susceptibility of mosquitoes to malaria infection. All mosquitoes used in the experiments were from the same population collected from the field in 2017 and since maintained under laboratory conditions. The experiments were carried out using different isolates of *P. relictum*. However, contrary to what might be expected, the isolates generating the highest oocyst burdens in mosquitoes had a similar parasitaemia and gametocytaemia to those generating lower oocyst burdens (electronic supplementary material, table S1). This suggests that parameters other than those measured in this study are responsible for the transmission rate or the development of *Plasmodium* in mosquitoes (e.g. sex ratio [78]). Interestingly, an effect of blood meal size on oocyst burden was observed, but only in some experiments. The positive relationship between blood meal size and the intensity of infection in mosquitoes seems to be mediated by the parasite density within the vertebrate host: the lower the parasite load within the vertebrate host, the larger the bloodmeal size effect seems to be.

In summary, we provide evidence that the overdispersion of parasite burden observed in mosquitoes fed on the same infected host is partly explained by a temporal heterogeneity in *Plasmodium* infectivity resulting from the biting order of mosquitoes. These results show that the parasite is either directly or indirectly capable of responding to the bites of mosquitoes to increase its own transmission over much shorter time scales than previously thought. Further work is required to elucidate whether these two strategies are complementary and, particularly, to elucidate the underlying mechanisms. Despite recent progress toward disease control, the number of malaria cases has increased in several countries. The efficacy of control strategies is continually challenged and threatened by the evolution of insecticide [79] and drug [80] resistances. To overcome these issues, the development of innovative therapeutic approaches is necessary and urgent. Understanding the mechanisms allowing *Plasmodium* to increase transmission in response to mosquito bites could lead to the development of new pharmaceutical approaches to control malaria transmission.

Ethics. This study was approved by the Ethical Committee of the Vaud Canton veterinary authority, authorization number 1730.4.

Data accessibility. All data supporting the conclusions of this paper are available on the Dryad Digital Repository: https://dx.doi.org/10.5061/dryad.ns1rn8pps [81].

Authors' contributions. A.R. and R.P. conceived the study and all authors elaborated the experimental design. R.P. and J.I. performed the experiments and analysed the data. R.P. and J.I. wrote the first draft of the manuscript, and all authors contributed substantially to revision.

Competing interests. We declare we have no competing interests.

Funding. This project was funded by the Swiss National Science Foundation (SNSF), grants 31003A_159600 and 31003A_179378 to P.C.

Acknowledgements. We would thank two anonymous reviewers for helpful and constructive comments. We would like to thank Tomas Kay for comments on the manuscript.

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
