## [Reviewer comments · Proceedings of the Royal Society B: Biological Sciences]

Review History

RSPB-2020-0917.R0 (Original submission)

Review form: Reviewer 1 (Jacques Charwood)

Recommendation

Reject – article is not of sufficient interest (we will consider a transfer to another journal)

Scientific importance: Is the manuscript an original and important contribution to its field?

Acceptable

General interest: Is the paper of sufficient general interest?

Acceptable

Quality of the paper: Is the overall quality of the paper suitable?

Acceptable

Is the length of the paper justified?

Yes

Should the paper be seen by a specialist statistical reviewer?

Yes

Do you have any concerns about statistical analyses in this paper? If so, please specify them explicitly in your report.

Yes

It is a condition of publication that authors make their supporting data, code and materials available - either as supplementary material or hosted in an external repository. Please rate, if applicable, the supporting data on the following criteria.

Is it accessible?

No

Is it clear?

No

Is it adequate?

No

Do you have any ethical concerns with this paper?

Yes

Comments to the Author

Please see my attached file for comments on your paper. (See Appendix A)

Review form: Reviewer 2 (Lisa Reimer)

Recommendation

Accept with minor revision (please list in comments)

Scientific importance: Is the manuscript an original and important contribution to its field?

Good

General interest: Is the paper of sufficient general interest?

Good

Quality of the paper: Is the overall quality of the paper suitable?

Good

Is the length of the paper justified?

Yes

Should the paper be seen by a specialist statistical reviewer?

No

Do you have any concerns about statistical analyses in this paper? If so, please specify them explicitly in your report.

No

It is a condition of publication that authors make their supporting data, code and materials available - either as supplementary material or hosted in an external repository. Please rate, if applicable, the supporting data on the following criteria.

Is it accessible?

No

Is it clear?

N/A

Is it adequate?

N/A

Do you have any ethical concerns with this paper?

No

Comments to the Author

The manuscript by Isaia and colleagues explores the poorly understood phenomenon of heterogeneity in mosquito infection intensities. They have designed elegant experiments to determine the temporal component of infection heterogeneity and provided strong evidence that the differences in intensity were not due to changes in the parasite burden in the peripheral or difference in mosquito bloodfeeding behaviours. This manuscript is well written and is an important contribution to our understanding of malaria transmission dynamics. We have identified a few minor but essential recommendations to improve the manuscript.

Introduction:

Line 32 – The use of the word “or” suggests that the two terms: aggregated and overdispersed are exclusive, suggest using an inclusive conjunction

Line 58 – No need to restrict the importance of malaria to the African continent since severe malaria occurs elsewhere as well

Line 64-65 – As written this sentence suggests sporozoites are produced in the salivary glands, suggest clearer language

Line 77 – Note that spatial aggregation does not explain the common observation in laboratory settings of widely different infection intensities

Line 158 – Does “unexposed birds” refer to birds that did not receive mosquito bites or birds that were not infected? It isn’t immediately clear what this is controlling for.

Line 207 – Typo – “experiment” rather than experience

Line 220 – Spell out likelihood ratio test since it is only used once

Results:

Multiple birds were exposed to mosquitoes at the same time. Is it possible there was preferential feeding on the different birds? i.e. did mosquitoes feed randomly on all the birds with no apparent host preference?

What were the initial levels of parasitemia in the birds? Consider the potential for preferential feeding on higher vs lower infected birds or specific feeding order on certain birds in different mosquito batches that might have affected oocyst burden.

Experiment 1-block-2 and Experiment 2: if quantity of haematin (proxy for blood meal size) was similar and it was not associated with biting order, how does it positively correlate with the increased oocyst intensity?

Please provide mean values on prevalence and hematin content rather than just the statistics on the model run. This is necessary to interpret the data and understand why burdens are so different between experiments.

Discuss why the oocyst burden is 10 fold different between the two experiments shown in Figure 2 A and B.

The number of parasites ingested does not necessarily correlate with the number of gametocytes

Discussion:

It would be worthwhile to note the proportion of variation is not explained by biting order to help guide further research.

Line 330 -What is the proposed mode of action for the fluctuation in plasmodium infectivity, is there any other evidence that suggests this is a host factor or a parasite factor?

Line 338 – React how? By stimulating growth and differentiation?

Line 341 Since prevalence is not significantly affected by biting order, is it appropriate to use the term ‘increased transmission’ to indicate an increase in the intensity of infection? The ability of

the parasite to be transmitted to the mosquito is not affected as such, but parasite development seems to be. As they discuss later, it may be that the increased infectivity has to do with improved survival in the mosquito. To me that would be 'improved development' rather than 'transmission' (?).

If they refer to potential transmission to humans, they do not address the correlation between number of oocysts and number of sporozoites in salivary glands, which cannot be assumed to be linear.

Lines 356-358 In experiment 3, the number of parasites ingested is quantified by qPCR therefore it is not known whether these are asexuals or gametocytes. While the total amount does not seem to change, the asexual/sexual ratio composition may change. In experiment 1 no changes were observed in parasitemia and gametocitemia composition assessed by microscopy, however experiment 3 may be different (different parasite isolate used).

Line 372-373 Could this be explored with in vitro studies using mosquito saliva?

Line 384 There is a suggestion of alteration in the concentration of blood components in the vertebrate host after multiple biting activity. How likely is this? Would this be a systemic response? It would be good to test how biting order affects infection intensity in laboratory SMFA, where the serum is inactivated.

Figures:

Figure 1 I would refer to the x axis using the time points rather than batches or maybe adding both.

Figure 2. The inset charts suggest there will be a very different value of k between the two experiments, can this be estimated and added for discussion?

Figure 2A Was it possible to accurately count up to 1500 oocysts, or are these estimates?

Figure 2B Y axis labelled as 'blood parasitemia' however what it is measured here is parasites ingested by the mosquito.

Decision letter (RSPB-2020-0917.R0)

05-Jun-2020

Dear Dr Pigeault:

I am writing to inform you that your manuscript RSPB-2020-0917 entitled "Last-come, best served? Mosquito biting order and Plasmodium transmission" has, in its current form, been rejected for publication in Proceedings B.

This action has been taken on the advice of referees, who have recommended that substantial revisions are necessary. With this in mind we would be happy to consider a resubmission, provided the comments of the referees are fully addressed. However please note that this is not a provisional acceptance.

1) A 'response to referees' document including details of how you have responded to the comments, and the adjustments you have made.

- 2) A clean copy of the manuscript and one with 'tracked changes' indicating your 'response to referees' comments document.
- 3) Line numbers in your main document.

Sincerely,
Dr Sasha Dall
mailto: proceedingsb@royalsociety.org

Associate Editor
Board Member: 1

Comments to Author:

Thank you for Submitting your manuscript "Last-come, best served? Mosquito biting order and Plasmodium transmission" to Proceedings B. I have now received two reviews and evaluated the manuscript myself. While we all find the topic interesting, a number of issues have been raised by the reviewers that should all be addressed. For example, reviewer 1 raises important points about whether the birds had been exposed to mosquitos prior to the experiments described here and how outliers may have influenced the results in experiment 1. Reviewer 2 highlights several technical and descriptive issues that will help improve the clarity of the manuscript.

Reviewer(s)' Comments to Author:

Referee: 1

Comments to the Author(s)

Please see my attached file for comments on your paper.

Referee: 2

Comments to the Author(s)

The manuscript by Isaïa and colleagues explores the poorly understood phenomenon of heterogeneity in mosquito infection intensities. They have designed elegant experiments to determine the temporal component of infection heterogeneity and provided strong evidence that the differences in intensity were not due to changes in the parasite burden in the peripheral or difference in mosquito bloodfeeding behaviours. This manuscript is well written and is an important contribution to our understanding of malaria transmission dynamics. We have identified a few minor but essential recommendations to improve the manuscript.

Introduction:

Line 32 - The use of the word "or" suggests that the two terms: aggregated and overdispersed are exclusive, suggest using an inclusive conjunction

Line 58 - No need to restrict the important of malaria to the African continent since severe malaria occurs elsewhere as well

Line 64-65 - As written this sentence suggest sporozoites are produced in the salivary glands, suggest clearer language

Line 77 - Note that spatial aggregation does not explain the common observation in laboratory settings of wildly different infection intensities

Line 158 - Does "unexposed birds" refer to birds that did not receive mosquito bites or birds that were not infected? It isn't immediately clear what this is controlling for.

Line 207 - Typo - "experiment" rather than experience

Line 220 - Spell out likelihood ratio test since it is only used once

Results:

Multiple birds were exposed to mosquitoes at the same time. Is it possible there was preferential feeding on the different birds? i.e. did mosquitoes feed randomly on all the birds with no apparent host preference?

What were the initial levels of parasitemia in the birds? Consider the potential for preferential feeding on higher vs lower infected birds or specific feeding order on certain birds in different mosquito batches that might have affected oocyst burden.

Experiment 1-block-2 and Experiment 2: if quantity of haematin (proxy for blood meal size) was similar and it was not associated with biting order, how does it positively correlate with the increased oocyst intensity?

Please provide mean values on prevalence and hematin content rather than just the statistics on the model run. This is necessary to interpret the data and understand why burdens are so different between experiments.

Discuss why the oocyst burden is 10 fold different between the two experiments shows in Figure 2 A and B.

The number of parasites ingested does not necessarily correlate with the number of gametocytes
Discussion:

It would be worthwhile to note the proportion of variation is not explained by biting order to help guide further research.

Line 330 -What is the proposed mode of action for the fluctuation in plasmodium infectivity, is there any other evidence that suggests this is a host factor or a parasite factor?

Line 338 – React how? By stimulating growth and differentiation?

Line 341 Since prevalence is not significantly affected by biting order, is it appropriate to use the term ‘increased transmission’ to indicate an increase in the intensity of infection? The ability of the parasite to be transmitted to the mosquito is not affected as such, but parasite development seems to be. As they discuss later, it may be that the increased infectivity has to do with improved survival in the mosquito. To me that would be ‘improved development’ rather than ‘transmission’ (?).

If they refer to potential transmission to humans, they do not address the correlation between number of oocysts and number of sporozoites in salivary glands, which cannot be assumed to be linear.

Lines 356-358 In experiment 3, the number of parasites ingested is quantified by qPCR therefore it is not known whether these are asexuals or gametocytes. While the total amount does not seem to change, the asexual/sexual ratio composition may change. In experiment 1 no changes were observed in parasitemia and gametocitemia composition assessed by microscopy, however experiment 3 may be different (different parasite isolate used).

Line 372-373 Could this be explored with in vitro studies using mosquito saliva?

Line 384 There is a suggestion of alteration in the concentration of blood components in the vertebrate host after multiple biting activity. How likely is this? Would this be a systemic response? It would be good to test how biting order affects infection intensity in laboratory SMFA, where the serum is inactivated.

Figures:

Figure 1 I would refer to the x axis using the time points rather than batches or maybe adding both.

Figure 2. The inset charts suggest there will be a very different value of k between the two experiments, can this be estimated and added for discussion?

Figure 2A Was it possible to accurately count up to 1500 oocysts, or are these estimates?

Figure 2B Y axis labelled as ‘blood parasitemia’ however what it is measured here is parasites ingested by the mosquito.

Author's Response to Decision Letter for (RSPB-2020-0917.R0)

See Appendix B.

RSPB-2020-2615.R0

Review form: Reviewer 1 (Jacques Charwood)

Recommendation

Accept as is

Scientific importance: Is the manuscript an original and important contribution to its field?

Good

General interest: Is the paper of sufficient general interest?

Good

Quality of the paper: Is the overall quality of the paper suitable?

Good

Is the length of the paper justified?

Yes

Should the paper be seen by a specialist statistical reviewer?

No

Do you have any concerns about statistical analyses in this paper? If so, please specify them explicitly in your report.

No

It is a condition of publication that authors make their supporting data, code and materials available - either as supplementary material or hosted in an external repository. Please rate, if applicable, the supporting data on the following criteria.

Is it accessible?

No

Is it clear?

No

Is it adequate?

No

Do you have any ethical concerns with this paper?

No

Comments to the Author

I wonder what your figures would look like on a log scale. Am I correct in assuming that some birds had more than 1000 oocysts? For Plasmodium falciparum oocyst rates are very much lower and a mosquito with more than 10 oocysts is heavily infected. Years ago some insects would be seen with 50+ oocysts but in recent years these insects are very few and far between.

Decision letter (RSPB-2020-2615.R0)

02-Nov-2020

Dear Dr Pigeault

I am pleased to inform you that your manuscript RSPB-2020-2615 entitled "Last-come, best served? Mosquito biting order and Plasmodium transmission" has been accepted for publication in Proceedings B.

The referee(s) have recommended publication, but also suggest some minor revisions to your manuscript. Therefore, I invite you to respond to the referee(s)' comments and revise your manuscript. Because the schedule for publication is very tight, it is a condition of publication that you submit the revised version of your manuscript within 7 days. If you do not think you will be able to meet this date please let us know.

[http://datadryad.org/submit?journalID=RSPB&manu=\(Document not available\)](http://datadryad.org/submit?journalID=RSPB&manu=(Document%20not%20available)) which will take you to your unique entry in the Dryad repository. If you have already submitted your data to dryad you can make any necessary revisions to your dataset by following the above link. Please see <https://royalsociety.org/journals/ethics-policies/data-sharing-mining/> for more details.

Sincerely,

Dr Sasha Dall

Associate Editor

Board Member

Comments to Author:

Thank you for addressing the review comments and updating the manuscript. The manuscript is much stronger now. One of the reviewers has made a couple of very minor additional comments that should be considered.

Reviewer(s)' Comments to Author:

Referee: 1

Comments to the Author(s).

I wonder what your figures would look like on a log scale. Am I correct in assuming that some birds had more than 1000 oocysts? For *Plasmodium falciparum* oocyst rates are very much lower and a mosquito with more than 10 oocysts is heavily infected. Years ago some insects would be seen with 50+ oocysts but in recent years these insects are very few and far between.

Author's Response to Decision Letter for (RSPB-2020-2615.R0)

See Appendix C.

Decision letter (RSPB-2020-2615.R1)

04-Nov-2020

Dear Dr Pigeault

I am pleased to inform you that your manuscript entitled "Last-come, best served? Mosquito biting order and Plasmodium transmission" has been accepted for publication in Proceedings B.

Open Access

Paper charges

Sincerely,
Editor, Proceedings B
<mailto:proceedingsb@royalsociety.org>

Appendix A

This is an interesting paper that, like all good work, tends to pose more questions than it answers. Some of these questions include such things as ‘What happens if feeding is actually started later in the night?’ ‘What happens if birds are injected with mosquito saliva prior to other insects feeding?’ ‘Would feeding by non-vectors (?Aedes) also impact the likelihood of a vector becoming infected?’ ‘Does the anti-body response of the host to mosquito saliva change in such a short period of time?’ (this may be important in a small host, like a canary, compared to a human who may need a much larger number of bites get it going), to name just a few.

Had the birds been exposed to mosquito bites before being used in the experiments? In other words, did they already perhaps have some anti-saliva antibodies that could be triggered by the feeding of the insects? I hope that they survive infection.

I note that the chi square value for haematin amount is considerably higher than that of feeding order (model 15) than any other, yet this is not mentioned in the discussion. Thus, feeding time is just one factor that may be involved – but it appears that it is a factor. I also wonder what happens in the analysis of experiment 1 if the outliers (the individual insects with a high oocyst burden in batch 4) are removed from the analysis. If the significance disappears under this scenario then the conclusion is that the likelihood of an occasional insect having a very high oocyst burden increases with feeding time rather than the population as a whole has a higher oocyst burden. I don’t know whether enough insects had such high burdens to rule out a random effect. How important that might be biologically is uncertain.

It is also moot whether the work can have any practical significance. This is largely because undertaking the experiments with human malaria is not likely to be possible due to ethical considerations. Thus, even if a novel control that incorporates this idea is proposed it is unlikely that it can be tested. Of greater relevance perhaps is the possibility that a change in biting time (if it is genetically based) among vectors as a response to wide-scale use of nets to the early evening reduces transmission to the mosquito. Again hard to test.

The English in the discussion needs some revision. For example:

Line 363 use was instead of has been

Line 366 a study carried out with the rodent

Line 370 the mature stage

(there are a number of other missing ‘the’s in some places and a few too many elsewhere)

Appendix B

UNIL | Université de Lausanne

19th of October 2020

Prof. Spencer Barret

Chief Editor, *Proceedings of the Royal Society B*.

Manuscript ID RSPB-2020-0917 “Last-come, best served? Mosquito biting order and *Plasmodium* transmission” (Isaïa et al.)

Dear Editor,

Many thanks for allowing us to resubmit our work to *Proceedings of the royal society B*. We are very grateful for the time you have spent evaluating our work and for providing constructive suggestions that have greatly improved our manuscript. We reply below to all the comments (our responses in bold) and we hope that you will now find our manuscript suitable for publication in *Proceedings of the Royal Society B*.

Yours sincerely,

Romain Pigeault, on behalf of all coauthors

Associated Editor

Thank you for Submitting your manuscript “Last-come, best served? Mosquito biting order and Plasmodium transmission” to Proceedings B. I have now received two reviews and evaluated the manuscript myself. While we all find the topic interesting, a number of issues have been raised by the reviewers that should all be addressed. For example, reviewer 1 raises important points about whether the birds had been exposed to mosquitos prior to the experiments described here and how outliers may have influenced the results in experiment 1. Reviewer 2 highlights several technical and descriptive issues that will help improve the clarity of the manuscript.

Thank you for the careful examination of our manuscript. Please find below a detailed, point-by-point, response to all the issues raised by yourself and the two reviewers.

Reviewer 1

This is an interesting paper that, like all good work, tends to pose more questions than it answers. Some of these questions include such things as ‘What happens if feeding is actually started later in the night?’ ‘What happens if birds are injected with mosquito saliva prior to other insects feeding?’ ‘Would feeding by non-vectors (?Aedes) also impact the likelihood of a vector becoming infected?’ ‘Does the anti-body response of the host to mosquito saliva change in such a short period of time?’ (this may be important in a small host, like a canary, compared to a human who may need a much larger number of bites get it going), to name just a few.

R1_1: We thank the reviewer for these positive comments and suggestions for future experiments. We agree that this study raises many interesting questions and we are currently exploring future avenues of research to expand these results

Had the birds been exposed to mosquito bites before being used in the experiments? In other words, did they already perhaps have some anti-saliva antibodies that could be triggered by the feeding of the insects? I hope that they survive infection.

R1_2: Birds were never exposed to mosquito before the experiments (they were infected by intra-peritoneal injection). We have now made this clear in the manuscript (see lines 128-129).

I note that the chi square value for haematin amount is considerably higher than that of feeding order (model 15) than any other, yet this is not mentioned in the discussion. Thus, feeding time is just one factor that may be involved – but it appears that it is a factor.

R1_3 In experiment 2 there was indeed a strong effect of blood meal size on the parasite burden. This effect, although weaker, is also observed in block 2 of experiment 1. The results of this and previous studies suggest that the relationship between bloodmeal size and oocyst burden in mosquitoes seem to be mediated by the parasitaemia in the bird (which is itself correlated with the gametocyaemia, See Pigeault et al.,2015): the lower the bird

parasite load, the larger the effect of the bloodmeal size on oocyst burden. This is now mentioned in the discussion (lines 436-440).

- Pigeault R, Vézilier J, Cornet S, Zélé F, Nicot A, Perret P, Gandon S, Rivero A. 2015 Avian malaria: a new lease of life for an old experimental model to study the evolutionary ecology of Plasmodium. *Phil Trans R Soc B* 370, 20140300. (doi:10.1098/rstb.2014.0300)

I also wonder what happens in the analysis of experiment 1 if the outliers (the individual insects with a high oocyst burden in batch 4) are removed from the analysis. If the significance disappears under this scenario then the conclusion is that the likelihood of an occasional insect having a very high oocyst burden increases with feeding time rather than the population as a whole has a higher oocyst burden. I don't know whether enough insects had such high burdens to rule out a random effect. How important that might be biologically is uncertain.

R1_4: This is an interesting point. Indeed, the more insects we test the higher the probability that one of them will have a very high oocyst number just by chance, although we do not see any particular reason why these random-only events should only happen at the end of the feeding bout. Following the reviewer's suggestion, we have re-ran the analyses by removing outliers (package grDevices, threshold: Q3 + 1.5*IR). The mosquito biting order still had a very significant effect on the oocyst burden (Experiment 1, block 1 : $\chi^2 = 12.431$, $p = 0.006042$, geometric mean without outliers - batch 1: 3.41 ± 3.04 , batch 2: 3.43 ± 2.82 , batch 3: 6.13 ± 3.36 , batch 4: 7.52 ± 2.82 , Experiment 1, block 2 : $\chi^2 = 19.307$, $p = 0.0002362$, geometric mean without outliers - batch 1: 3.48 ± 2.69 , batch 2: 3.26 ± 2.77 , batch 3: 5.63 ± 2.72 , batch 4: 8.13 ± 2.39). This is now mentioned in the results section (See lines 244-248 and 271-274).

It is also moot whether the work can have any practical significance. This is largely because undertaking the experiments with human malaria is not likely to be possible due to ethical considerations. Thus, even if a novel control that incorporates this idea is proposed it is unlikely that it can be tested. Of greater relevance perhaps is the possibility that a change in biting time (if it is genetically based) among vectors as a response to wide-scale use of nets to the early evening reduces transmission to the mosquito. Again hard to test.

R1_5: In our view, understanding the mechanisms allowing parasites to increase transmission from vertebrate to invertebrate hosts in response to mosquito bites could lead to the development of new approaches to reduce parasite burden in the vectors and its effects on the probability of transmission (Churcher et al 2017). For instance, if the effect observed is the result of a plastic switch in the parasite in response to mosquito saliva, tools that turn off the molecular switch could be developed. To transpose these experiments to human malaria, one possibility would be to carry out in vitro experiments quantifying e.g. gametocyte conversion, maturation rates or exflagellation rates in the presence of serum from humans bitten or unbiten by uninfected mosquitoes. Another interesting offshoot of our results is that we may expect the degree of overdispersion to depend on the mean biting

rate (number of biting rates per human per night) of mosquitoes in a given area. These points have now been mentioned in the discussion (lines 409-424).

- Churcher TS *et al.* 2017 Probability of transmission of malaria from mosquito to human is regulated by mosquito parasite density in naïve and vaccinated hosts. *PLOS Pathog.* 13, e1006108. (doi:10.1371/journal.ppat.1006108)

The English in the discussion needs some revision. For example:

Line 363 use was instead of has been

Line 366 a study carried out with the rodent

Line 370 the mature stage

(there are a number of other missing 'the's in some places and a few too many elsewhere)

R1_6: Thank you, we corrected these issues and we had it proofread by English speakers.

Reviewer 2

The manuscript by Isaïa and colleagues explores the poorly understood phenomenon of heterogeneity in mosquito infection intensities. They have designed elegant experiments to determine the temporal component of infection heterogeneity and provided strong evidence that the differences in intensity were not due to changes in the parasite burden in the peripheral or difference in mosquito blood feeding behaviours. This manuscript is well written and is an important contribution to our understanding of malaria transmission dynamics. We have identified a few minor but essential recommendations to improve the manuscript.

Introduction

Line 32 – The use of the word “or” suggests that the two terms: aggregated and overdispersed are exclusive, suggest using an inclusive conjunction

R2_1: Thank you. To make this sentence clearer we now only use the term overdispersed.

Line 58 – No need to restrict the important of malaria to the African continent since severe malaria occurs elsewhere as well

R2_2: Corrected, you are absolutely right.

Line 64-65 – As written this sentence suggest sporozoites are produced in the salivary glands, suggest clearer language

R2_3: Thank you, we modified this sentence (see lines 62-63).

Line 77 – Note that spatial aggregation does not explain the common observation in laboratory settings of wildly different infection intensities

R2_4: Although in the lab only certain parts of the infected host (bird, mouse) is presented to the mosquito, there's still room for spatial aggregation of the infective stages. Indeed, in a recent study (Pigeault et al., 2020) we have observed that gametocytes are not homogeneously distributed within the vertebrate host and vectors fed on the least infected body part had a lower parasite burden than those fed on the most infected part. We have nevertheless modified this sentence in order to nuance our statements (see lines 74-77).

- Pigeault R, Isaïa J, Yerbanga RS, Dabiré KR, Ouédraogo J-B, Cohuet A, Lefèvre T, Christe P. 2020 Different distribution of malaria parasite in left and right extremities of vertebrate hosts translates into differences in parasite transmission. *Sci. Rep.* 10, 10183. (doi:10.1038/s41598-020-67180-6)

Line 158 – Does “unexposed birds” refer to birds that did not receive mosquito bites or birds that were not infected? It isn't immediately clear what this is controlling for.

R2_5: We agree, we modified this sentence (see line 158). Unexposed birds refer to birds unexposed to mosquito bites.

Line 207 – Typo – “experiment” rather than experience

R2_6: Thank you.

Line 220 – Spell out likelihood ratio test since it is only used once

R2_7: Done, thank you.

Results

Multiple birds were exposed to mosquitoes at the same time. Is it possible there was preferential feeding on the different birds? i.e. did mosquitoes feed randomly on all the birds with no apparent host preference?

R2_8: We're afraid that the wording of the manuscript may have led to a misunderstanding. Each bird was placed individually in an experimental cage and the mosquitoes were then added to each cage (no bird choice was involved). We explain this more clearly in the material and method (see lines 149 and 176).

What were the initial levels of parasitemia in the birds? Consider the potential for preferential feeding on higher vs lower infected birds or specific feeding order on certain birds in different mosquito batches that might have affected oocyst burden.

R2_9: Please see above for the clarification on the lack of bird choice by the mosquitoes. The initial levels of bird parasitaemia were highly variable (mean \pm SE: 3.11% \pm 0.82, range: 0.01 – 17.22). We have added a table in supplementary information with the parasitaemia and gametocyaemia of birds before and after mosquito exposure (see Table S1). As mentioned in our manuscript we did not observed any effect of mosquito batches (feeding

order) on the proportion of blood-fed mosquitoes (blood meal rate). We have pooled the results of all the experiments and analyzed the impact of bird parasitaemia (before mosquito exposure) on the percentage mosquitoes taking a blood and the effect was non-significant ($F = 0.4873$, $p = 0.5113$). Although our experiments are not designed to test this, there does not appear to be a preferential feeding on higher vs lower infected birds.

Experiment 1-block-2 and Experiment 2: if quantity of haematin (proxy for blood meal size) was similar and it was not associated with biting order, how does it positively correlate with the increased oocyst intensity?

R2_10: This can be explained by the fact that despite a significant increase in oocyst burden according to mosquito biting order, there was also fluctuations in parasite density between mosquitoes belonging to the same batch (Experiment 1) or between mosquitoes that have taken their blood meal in close time intervals (Experiment 2 & 3). There thus appear to be at least two independent parameters that induce an increase in oocyst burden i. bloodmeal size ii. the order of mosquito bites.

Please provide mean values on prevalence and hematin content rather than just the statistics on the model run.

R2_11: We now added all mean values on blood meal rate, infection prevalence and excreted haematin.

This is necessary to interpret the data and understand why burdens are so different between experiments. Discuss why the oocyst burden is 10-fold different between the two experiments shows in Figure 2 A and B.

R2_12: We agree. We now discuss this point (see lines 425-435).

The number of parasites ingested does not necessarily correlate with the number of gametocytes

R2_13: In the avian malaria system (and especially with *Plasmodium relictum* parasite) there is a strong positive correlation between parasitaemia and the number of gametocytes (see Figure 2 in Pigeault et al 2015). This positive correlation was also observed here ($df = 54$, $F = 175$, $p < 0.001$, see also Table S1). We have added this information in lines 135-138.

- Pigeault R, Vézilier J, Cornet S, Zélé F, Nicot A, Perret P, Gandon S, Rivero A. 2015 Avian malaria: a new lease of life for an old experimental model to study the evolutionary ecology of Plasmodium. Phil Trans R Soc B 370, 20140300. (doi:10.1098/rstb.2014.0300)

Discussion

It would be worthwhile to note the proportion of variation is not explained by biting order to help guide further research.

R2_14: We agree, we added the average proportion of variation in oocyst burden explained by mosquito biting order (10% \pm 3, lines 340-341).

Line 330 -What is the proposed mode of action for the fluctuation in plasmodium infectivity, is there any other evidence that suggests this is a host factor or a parasite factor?

R2_15: Previous studies have shown that vertebrate host parasitaemia increases a few days after a mosquito blood feeding bout, suggesting that *Plasmodium* may be capable of adjusting its transmission strategy by responding plastically to mosquito bites (Cornet 2014, Pigeault 2018). Here, within a short period of time (3h), we did not observe any variation in parasite burden before and after mosquito exposure. We discussed alternative mechanisms involving an active response of the parasite (e.g. variation in the rate of gametocyte maturation) but also responses of the vertebrate host to cumulative mosquito bites (e.g., antibodies, complement components).

Line 338 – React how? By stimulating growth and differentiation?

R2_16: Previous studies showed that *Plasmodium relictum* and *Plasmodium chabaudi* react to mosquito bites by increasing their overall parasitaemia in the vertebrate blood. However, there was no evidence of any change in the conversion rate (density of gametocytes relative to the total number of parasites, Billingsley et al 2005, Cornet et al 2014, Pigeault et al 2018). We modified this sentence (lines 351-352).

- Pigeault R, Caudron Q, Nicot A, Rivero A, Gandon S. 2018 Timing malaria transmission with mosquito fluctuations. *Evol. Lett.* (doi:10.1002/evl3.61)

- Cornet S, Nicot A, Rivero A, Gandon S. 2014 Evolution of plastic transmission strategies in avian malaria. *PLoS Pathog* 10, e1004308. (doi:10.1371/journal.ppat.1004308)

- Billingsley PF, Snook LS, Johnston VJ. 2005 Malaria parasite growth is stimulated by mosquito probing. *Biol. Lett.* 1, 185–189. (doi:10.1098/rsbl.2004.0260)

Line 341 Since prevalence is not significantly affected by biting order, is it appropriate to use the term 'increased transmission' to indicate an increase in the intensity of infection? The ability of the parasite to be transmitted to the mosquito is not affected as such, but parasite development seems to be. As they discuss later, it may be that the increased infectivity has to do with improved survival in the mosquito. To me that would be 'improved development' rather than 'transmission' (?). If they refer to potential transmission to humans, they do not address the correlation between number of oocysts and number of sporozoites in salivary glands, which cannot be assumed to be linear.

R2_17: Thank you for this relevant comment. We have indeed shown in the experiment 3 that the amount of parasite ingested by mosquitoes did not vary with the mosquito biting order so we cannot actually talk about an increase in transmission. We have therefore replaced "transmission" by "parasite burden" and "intensity of infection". Concerning the "development" of parasites in mosquitoes, given the structure of our discussion, we believe

that it is preferable to discuss the mechanisms that can lead to an increase in parasite burden in the subsequent paragraphs.

Lines 356-358 In experiment 3, the number of parasites ingested is quantified by qPCR therefore it is not known whether these are asexuals or gametocytes. While the total amount does not seem to change, the asexual/sexual ratio composition may change. In experiment 1 no changes were observed in parasitemia and gametocitemia composition assessed by microscopy, however experiment 3 may be different (different parasite isolate used).

R2_18: We do not have molecular tools to discriminate sexual and asexual stages. Nevertheless Experiment 3 was carried out with the same parasite isolate used in the block 2 of the first experiment. The parasite was only passaged once by intraperitoneal (i.p.) infection between the two experiments. Please note also that in this system, total parasitaemia is a good predictor of gametocytaemia (see previous reply R2_13)

Line 372-373 Could this be explored with in vitro studies using mosquito saliva?

R2_19: That's right. Please see reply R1_5

Line 384 There is a suggestion of alteration in the concentration of blood components in the vertebrate host after multiple biting activity. How likely is this? Would this be a systemic response? It would be good to test how biting order affects infection intensity in laboratory SMFA, where the serum is inactivated.

R2_20:

- In addition to the trauma caused by the insertion of the mosquito proboscis into the skin, a cutaneous reaction to mosquito bites occur due to effects of saliva (localized response). Mosquito saliva is a complex mixture of biologically active factors that induce dermal inflammation (e.g. mosquito bites influence the cutaneous microenvironment by inducing a differential migration of immune cells, Demeure *et al.*, 2005, Pingen *et al.*, 2017, Henrique *et al.*, 2019). A systemic reaction may be also observed with, for instance, an increase in stress hormone concentration. We started a study where we investigated the impact of mosquito bites on corticosterone production in birds and our preliminary results showed that exposure to mosquitoes increases blood corticosterone concentration by 2.5 times. Interestingly some studies have suggested that stress-related hormonal changes would increase hematozoan parasitemia (Dhondt & Dobson 2017).

Here, we suggest that, if present, mosquito-triggered reactions in the blood may increase with the number of mosquito bites.

- We disagree that a membrane-feeding assay would be able to provide any clarity on any systemic response triggered by the mosquito bites; as if present, the response is likely to involve a whole cascade of metabolic/immune/hormonal pathways whose sources are not necessarily in the blood.

- We added a paragraph at the end of the discussion where we develop the different studies that could be conducted to go further to explore these issues (see lines 409-424).

- Demeure CE et al. 2005 Anopheles mosquito bites activate cutaneous mast cells leading to a local inflammatory response and lymph node hyperplasia. *J. Immunol. Baltim. Md* 174, 3932–3940. (doi:10.4049/jimmunol.174.7.3932)
- Pingen M, Schmid MA, Harris E, McKimmie CS. 2017 Mosquito Biting Modulates Skin Response to Virus Infection. *Trends Parasitol.* 33, 645–657. (doi:10.1016/j.pt.2017.04.003)
- Dhondt AA, Dobson AP. 2017 Stress Hormones Bring Birds, Pathogens and Mosquitoes Together. *Trends Parasitol.* 33, 339–341. (doi:10.1016/j.pt.2017.01.001)
- Pigeault R, Isaïa J, Yerbanga RS, Dabiré KR, Ouédraogo J-B, Cohuet A, Lefèvre T, Christe P. 2020 Different distribution of malaria parasite in left and right extremities of vertebrate hosts translates into differences in parasite transmission. *Sci. Rep.* 10, 10183. (doi:10.1038/s41598-020-67180-6)

Figures

Figure 1 I would refer to the x axis using the time points rather than batches or maybe adding both.

R2_21: Done, thank you

Figure 2. The inset charts suggest there will be a very different value of k between the two experiments, can this be estimated and added for discussion?

R2_22: We now estimated and added the values of parameter k for each experiment (Clumping parameter $k = \mu^2 / \sigma^2 - \mu$). Interestingly, the range of values (range: 0.35 - 1.5) correspond to those found in previous experiments performed with other malaria systems (Pichon et al. 2000, Gaillard et al. 2003).

- Pichon G, Awono-Ambene HP, Robert V. 2000 High heterogeneity in the number of *Plasmodium falciparum* gametocytes in the bloodmeal of mosquitoes fed on the same host. *Parasitology* 121 (Pt 2), 115–120.

- Gaillard FO, Boudin C, Chau NP, Robert V, Pichon G. 2003 Togetherness among *Plasmodium falciparum* gametocytes: interpretation through simulation and consequences for malaria transmission. *Parasitology* 127, 427–435.

Figure 2A Was it possible to accurately count up to 1500 oocysts, or are these estimates?

R2_23: We attempt as far as possible to count all the oocysts (we frequently spend more than 20 minutes/stomach). However, we cannot exclude that when the oocyst burden is very high (especially when the density exceeds 1000 oocyst/stomach) it is possible to miss some parasites and the counting becomes approximate.

Figure 2B Y axis labelled as ‘blood parasitemia’ however what it is measured here is parasites ingested by the mosquito.

R2_24: Done, thank you

Appendix C

UNIL | Université de Lausanne

4th of November 2020

Prof. Spencer Barret

Chief Editor, *Proceedings of the Royal Society B*.

Manuscript ID RSPB-2020-2615 “Last-come, best served? Mosquito biting order and *Plasmodium* transmission” (Isaïa et al.)

Dear Editor,

Thank you for accepting our manuscript for publication in *Proceedings of the Royal Society B*.

Please find enclosed our responses to the comments raised by Reviewer 1 and the revised version of our manuscript following the last round of reviews.

Yours sincerely,

Romain Pigeault, on behalf of all coauthors

Associated Editor

Thank you for addressing the review comments and updating the manuscript. The manuscript is much stronger now. One of the reviewers has made a couple of very minor additional comments that should be considered.

Thank you for the careful review of our manuscript. Please find below our responses to the comments raised by Reviewer 1.

Reviewer 1

I wonder what your figures would look like on a log scale.

R1_1: Our models are built to analyze data with a negative binomial distribution (glmer.nb), we think that showing raw data (the ones used in the models) in the figures is more relevant than showing transformed data that does not match what we have analyzed.

However please find below the figures with a log transformation, but it seems to us more relevant to keep the figures with the raw data in the manuscript.

Am I correct in assuming that some birds had more than 1000 oocysts? For *Plasmodium falciparum* oocyst rates are very much lower and a mosquito with more than 10 oocysts is heavily infected. Years ago some insects would be seen with 50+ oocysts but in recent years these insects are very few and far between.

R1_2: Indeed, the oocyst burden we find in the lab is way higher than what is found in the field in *Plasmodium falciparum*. All experimental infections carried out in the laboratory (e.g. human, rodent, avian) produce oocystaemias that are substantially larger than what we find in the field because infections are carried out under optimal conditions (blood meal on acute infection stage, no host defensive behavior, optimal temperature, humidity etc). In addition, given the negative correlation between oocyst burden and longevity (Dawes et al 2006) it is likely that heavily infected mosquitoes would not survive in the field. We now mention this point (see lines 401-405).

- Dawes EJ, Churcher TS, Zhuang S, Sinden RE, Basáñez M-G. 2009 Anopheles mortality is both age- and *Plasmodium*-density dependent: implications for malaria transmission. *Malar. J.* 8, 228. (doi:10.1186/1475-2875-8-228)

Figure 1: Experiment 1: Impact of mosquito batch order on *Plasmodium* transmission.

Number of oocysts (log scale) in the midgut of *Plasmodium*-infected mosquitoes according to mosquito batch. Each mosquito batch was left in contact with birds for 45 minutes (batch 1 ($T_{0_{min}}$), batch 2 ($T_{45_{min}}$), batch 3 ($T_{90_{min}}$) and batch 4 ($T_{135_{min}}$)). Birds were either infected by a *Plasmodium relictum* lab strain (experimental block 1, panel **A**) or by a *Plasmodium relictum* strain freshly collected in the field (experimental block 2, panel **B**). Black horizontal lines represent medians and black diamonds represent geometric means. Levels not connected by the same letter are significantly different. Histograms in each panel show the distribution of oocyst burden in mosquitoes in the experimental blocks 1 (**A**) and 2 (**B**), the colors represent the mosquito batches (from 1 to 4).

Figure 2: Effect of individual mosquito blood feeding order on the number of parasites ingested and on the intensity of infection. (A) Relationship between oocyst burden (log scale) and mosquito biting order (experiment 2). **(B)** Relationship between the number of parasites ingested ($\text{Log}(\text{RQ}+1)$, in red), or the oocyst burden (log scale, in black), and the mosquito biting order (experiment 3). Each point represents one blood-fed mosquito. Shaded areas on either side of the regression line represent 95% confidence intervals. Histograms show the distribution of oocyst burden in mosquitoes in the experiment 2 (panel A) and 3 (panel B).